# On the Problem of Restoring and Classifying a 3D Object in Creating a Simulator of a Realistic Urban Environment

**DOI:** 10.3390/s22145199

**Published:** 2022-07-12

**Authors:** Mikhail Gorodnichev, Sergey Erokhin, Ksenia Polyantseva, Marina Moseva

**Affiliations:** Faculty of Information Technology, Moscow Technical University of Communications and Informatics, 111024 Moscow, Russia; m.g.gorodnichev@mtuci.ru (M.G.); esd@mtuci.ru (S.E.); m.s.moseva@mtuci.ru (M.M.)

**Keywords:** artificial intelligence, neural networks, CNN, recognition

## Abstract

Since the 20th century, a rapid process of motorization has begun. The main goal of researchers, engineers and technology companies is to increase the safety and optimality of the movement of vehicles, as well as to reduce the environmental damage caused by the automotive industry. The difficulty of managing traffic flows is that cars are driven by a person and their behavior, even in similar situations, is different and difficult to predict. To solve this problem, ground-based unmanned vehicles are increasingly being developed and implemented; however, like any other intelligent system, it is necessary to train different road scenarios. Currently, an engineer is driving an unmanned vehicle for training and thousands of kilometers are being driven for training. Of course, this approach to training unmanned vehicles is very long, and it is impossible to reproduce all the scenarios that can be found in real operations on a real road. Based on this, we offer a simulator of a realistic urban environment which allows you to reduce the training time and allows you to generate all kinds of events. To implement such a simulator, it is necessary to develop a method that would allow recreating a realistic world in one passage with cameras (monocular) installed on board the vehicle. Based on this, the purpose of this work is to develop an intelligent vehicle recognition system using convolutional neural networks, which allows you to create mesh objects for further placement in the simulator. It is important to note that the resulting objects should be optimal in size so as not to overload the system, since a large number of road infrastructure objects are stored there. Also, neural complexity should not be excessive. In this paper, the general concept and classification of convolutional neural networks are given, which allow solving the problem of recognizing 3D objects in images. Based on the analysis, the existing neural network architectures do not solve the problems mentioned above. In this connection, the authors first of all carried out the design of the system according to the methodology of modeling business processes, and also modified and developed the architecture of the neural network, which allows classifying objects with sufficient accuracy, obtaining optimized mesh objects and reducing computational complexity. The methods proposed in this paper are used in a simulator of a realistic urban environment, which reduces the time and computational costs when training unmanned transport systems.

## 1. Introduction

Currently, in many large cities, the possibilities of developing transport networks are close to exhaustion, and car traffic is growing every year. In the current situation, it is necessary not only to design new roads qualitatively, but also to ensure the efficiency of their functioning and traffic safety. Solving these problems is impossible without mathematical modeling of transport networks, which allows you to determine parameters such as traffic intensity, average speed, delays and time loss.

In recent years, many mathematical models have been developed; however, these models do not work without real data, since a person with a high degree of uncertainty of actions is driving vehicles. Autonomous vehicles are being developed and put into operation to solve this problem. New solutions for the transport industry are being applied in a wide variety of areas, which allows to reduce energy costs and improve the ecology of cities by optimizing the movement of autonomous vehicles. Global technology giants such as Google, Uber, and Yandex are fighting to improve the safety of transportation; they are testing autonomous transport technologies in different regions of the world.

Unfortunately, at the moment, the process of rapid introduction of autonomous vehicles is hindered by 2 factors: people’s distrust of artificial intelligence and the great complexity of training autonomous driving systems. This work is aimed at reducing the impact of these factors by creating a simulation of a realistic urban environment, improving neural network detection/recognition algorithms in conditions of data uncertainty and developing effective optimization methods. It is important to note that for modern traffic planning of autonomous vehicles, it is necessary to consider the street and road network as a whole, and not on individual sections, as is being done now. In addition, the analysis of autonomous vehicle traffic control systems revealed a large number of problems faced by designers in the process of their development and in determining the requirements for the control system; this is due to the following objective factors: a sufficiently high error, the inability of most systems to take into account the constantly changing external conditions during movement, the functional limitations of control systems due to the use of external information sources that determine the position of the vehicle in space.

The aim of this article is to develop an intelligent system for vehicle recognition using convolutional neural networks.

Many scientists are engaged in research in the development of route planning methods: Abdul Wahid, Md Tanwir Uddin Haider, Md Masood Ahmad, Arvind Kumar Singh, Xinwu Qian, Jiawexue, etc. Ukkusuri Wojciech Chmiel, Iwona Skalna and Stanisław Jędrusik within the framework of the INSIGMA [1] project are investigating the use of a method based on interval numbers for traffic management in the city. Using this method and information about the current state of traffic, the proposed system helps to navigate more effectively in the changing environment of the city. The methodology is based on the continuous flow of information from the streets and prompt response to changes. Since traffic parameters are subject to fluctuations (including due to measurement accuracy), a mechanism has been proposed to prevent frequent route changes. Important results of the conducted research are the introduction of interval algebra into urban traffic management and the provision of a theoretical basis for determining conditions that increase the likelihood of obtaining the optimal path with traffic fluctuations; however, the disadvantages of existing systems are the lack of optimization of reward functions and sufficient convergence of the network to work in real-time.

Nohel et al. [2] describe the structure of the optimal route model implemented in TDSS, and its further possible use in the maneuver management system. In their calculations, they combine an assessment of the entire terrain and the safety characteristics of the working area; they focus on the cross-country capabilities of wheeled vehicles in terrain conditions; however, in the case of autonomous vehicles, there is still a need for direct control on the ground due to the accidental occurrence of microrelief forms and obstacles.

Aguiar et al. [3] describe PRONTO—a flexible and efficient numerical tool for exploring the space of one or more vehicles. The constraints of the state and the input inequality are processed using an approximate logarithmic barrier, which allows starting the iterative process from an impracticable trajectory. Current research efforts are aimed at using the model being developed as a reactive scheduler, and not just as an operational scheduler. As disadvantages of existing models, it is worth highlighting the lack of cost minimization when working in real mode.

Dario et al. [4,5] consider strategies for designing state-feedback control to stabilize the vehicle during maneuvers. Vehicle stabilization is achieved through a combination of steering, acceleration and braking. A model with linear parameter variation is obtained as a result of linearization of a nonlinear model along a reference trajectory. For the model, the control law with feedback on the state is calculated.

Peng et al. [6] investigates the joint time-varying maneuvering task of forming and maintaining connectivity, collision avoidance for autonomous ground vehicles with direction measurements. Time-varying joint maneuvering control laws are being developed based on artificial potential functions, nonlinear tracking differentiators and the reverse step method. The stability of a closed-loop distributed formation control system is analyzed on the basis of I/O stability and cascade stability. The disadvantages of existing developments are the lack of dynamic models capable of working in real time, while ensuring the proper level of safety of autonomous vehicles.

Frame synchronization is more difficult when moving objects or moving the camera sharply, because these factors lead to shaking, shakiness, defects and blurring. Image stitching solutions based on neural network algorithms are rarely studied due to the lack of ready-made sets of labeled data. Nie et al. [7] offer an image stitching algorithm based on unsupervised learning.

Gu et al. [8] explore the “gluing” of multiple images to realize the full spectrum of a global view of the internal environment. The research is devoted to demonstrating a large field of view using dynamic image stitching when there is a moving object in the environment.

Kulawik et al. [9,10] also use neural network algorithms based on convolutional neural networks (CNN), with which they track synchronization errors during the transmission of digital images. Previously, the synchronization problem was solved with the help of trigger triggers in the recording; this solution checks the discrepancy between the received pairs of images, which allows for detecting delays in the transmission of images between cameras. For this purpose, a deep network is used to classify the analyzed images into five classes.

However, the existing synchronization algorithms are not sufficient, more efficient and reliable algorithms are required since numerous studies have shown that the real data sets that need to be stitched are more complex than the test ones. One of the main criteria for checking an autonomous vehicle is safe interaction with other road users. Based on research, testing on real roads is sometimes impractical for safety testing due to their time and financial costs. Therefore, modeling the “traveled” kilometers is the only possible way to overcome this limitation. The development of methods for creating a simulation environment allows experiments to be carried out in a digital environment, rather than in a real one. The 3D scene of the road network helps to model the distribution of road infrastructure and the corresponding road conditions; however, the existing methods of modeling the movement of vehicles have limitations such as inflexibility in various types of modification of road infrastructure, poor quality of visual effects and low efficiency in rendering large-scale models, etc. To solve these problems, the method of 3D modeling of roads based on templates is often proposed; in such methods, road infrastructure data are first pre-processed before modeling. The centerlines of the roads are analyzed to extract information about the topology and recalculated to improve the accuracy of the trajectory and match the terrain. The following companies are engaged in developments in this area: Zhang et al. [11,12,13], Malayjerdi et al. [14].

However, it is important to note that the data of the road geoinformation system do not provide the complete information necessary to create 3D models of roads in complex cases. For example, without accurate information about the height, it is difficult to determine the location of various fences. Real-time 3D reconstruction is one of the currently popular areas of computer vision research; this task has become the main technology in the field of virtual reality, industrial automatic systems and trajectory planning of mobile robots. Currently, there are three main problems in the field of real-time 3D reconstruction. Firstly, it is expensive; this requires more diverse sensors, so it’s less convenient. Secondly, the recovery rate is low, and the 3D model cannot be accurately set in real time. Thirdly, the recovery error is large, which cannot accurately meet the requirements of the scenes.

Jia et al. [15] proposed a real-time 3D reconstruction method based on monocular vision. One RGB-D camera is used to collect visual information in real time, and the YOLACT++ network is used to identify and segment visual information to extract some of the important visual information. The three stages of depth recovery, depth optimization, and deep fusion are then combined to offer a three-dimensional position estimation method based on deep learning for co-coding visual information; this can reduce the depth error caused by the depth measurement process, and the exact values of the 3D points of the segmented image can be obtained directly. Then, a method based on a limited correction of the distance to the cluster center emissions is proposed to optimize the three-dimensional point values obtained above; this increases the accuracy of real-time reconstruction and allows you to get a three-dimensional model of the object in real time.

Sun et al. [16] presented a platform called NeuralRecon for real-time reconstruction of a 3D scene using monocular video. Unlike existing methods that evaluated depth maps with one view separately for each keyframe and combined them later, they propose to directly reconstruct local surfaces represented as sparse TSDF volumes for each video fragment sequentially using a neural network. The learning-based TSDF merge module, based on closed repeating blocks, is used to manage the network to combine functions from previous fragments; this sign allows the network to capture the local smoothness up to and the global shape up to 3D surfaces in sequential reconstruction of surfaces, resulting in accurate, consistent reconstruction of the surface in real time.

As a rule, existing research on semantic mapping represented approaches based on the use of cameras that could not be used in large-scale environments due to their computational load. Recently, a method of combining 3D lidar with a camera was introduced to solve this problem, and 3D lidar and camera were also used for semantic 3D mapping. In this study by Jeong et al. [17], the algorithm consists of semantic mapping and map refinement. In the semantic mapping, GPS and IMU are integrated to evaluate the odometry of the system, and subsequently point clouds measured using 3D lidar are recorded using this information. In addition, semantic segmentation based on CNN is used to obtain semantic information about the environment. To integrate the point cloud with semantic information, incremental semantic labeling has been developed, including coordinate alignment, error minimization, and semantic information integration. In addition, to improve the quality of the generated semantic map, the map refinement is processed in batch mode. As a disadvantage, it should be noted the low speed of the method.

Training models with high performance require a large set of marked-up data, the acquisition of which is resource-intensive. The aim of the work, which was carried out by Kar et al. [18], was the synthesis of labeled data sets that can be used for specific purposes; they parametrize the dataset generator with a neural network that learns to modify the attributes of scene graphs derived from probabilistic stage grammars in order to minimize the gap in distribution between the output and target data obtained. If the real data set comes with a small labeled test set, then researchers additionally strive to optimize the meta-goal, i.e., the final results of the task. As a disadvantage, it should be noted that there is no stage of post-processing by a neural network to bring the picture to the view of the real world.

The work consists of four parts. The first part is “Introduction”, which describes the relevance and problems that are solved in the article, as well as an analysis of existing neural network architectures aimed at solving the problem, identified shortcomings that are solved by the authors in the works. In the second part “Materials and Methods” the methods are considered, and the substantiation of the methods used by the authors is carried out. In the third part “Results”, the system was designed according to the methodology of business processes, the development and optimization of the architecture of the proposed neural network is described, and a comparison with analogues is made. The fourth part of the “Conclusion” presents the generalized results of the study and the possibility of application in real systems.

## 2. Materials and Methods

### 2.1. General Concept and Classification of Neural Networks

Here, we list the main advantages of neural methods in comparison with traditional methods.

Solving the problem of unknown patterns.

Traditional expert systems are not able to learn and gain new knowledge. At the same time, neural networks provide the output of new patterns between input and output data, which makes it possible to work on data that was not included in the training sample.

2.There is no guarantee of repetition and unambiguity of the final results.

However, in the field of knowledge representation and processing, neural networks also have advantages:Formalization of knowledge is not necessary; it can be replaced by learning by examples;Naturalness of processing and presentation of fuzzy knowledge, similar to the implementation in the brain;Parallel processing with proper hardware support creates conditions for real-time operation;Hardware implementation is able to provide fault tolerance;Processing of multidimensional data (more than three) without increasing labor intensity, as well as knowledge [19].

### 2.2. The Task of Recognizing 3D Objects in an Image

The task of reliable detection of objects in a three-dimensional scene has become relevant with the advent and development of methods for obtaining three-dimensional digital images. Meanwhile, sensors such as LiDAR and RGBD cameras have evolved and become an increasingly common solution for many autonomous robotics; this work is a continuation of research on methods for recognizing three-dimensional objects in a point cloud—a specialized format for representing three-dimensional data.

The methods considered by the authors in [19,20] effectively recognize objects, and their speed is sufficient for functioning in real time, but the main disadvantage of the described methods is the requirement of the identity of the desired object to a predetermined standard. That is, in order to recognize any registered object, it is necessary that the system already has a three-dimensional model associated with a certain class; this approach is well suited for recognizing objects on model data, but in reality, the registered object often differs from its representation in reference models. Recently, methods based on deep learning technology have become increasingly widespread.

One of the examples of the classical approach to the detection of a three-dimensional object is the three-dimensional Hough transform [21,22]; this method has become popular in application for two-dimensional images, where it mainly uses the contours of objects as areas of interest. When working in a three-dimensional format, the method allocates special points in a three-dimensional image to reduce computational costs. Such points are allocated using a special clustering algorithm. The voting procedure takes place in the accumulator space, taking into account only the selected special points. As a result, local maxima are obtained in those areas where the desired object can potentially be located. Another difficulty with adding a third dimension, in addition to the increased computational load—is the probability of different orientations of the scene and the desired object; this problem is solved by introducing special vectors that ensure invariance to rotation and rotation. A little more elaborate is the geometric connectivity method; the main difference from the three-dimensional Hough transform is a different algorithm for searching for singular points, which are combined into so-called special areas and translated into a format described by a special index of forms. The obtained areas in the form of index values are recorded in two-dimensional histograms, where the voting procedure takes place for all local neighborhoods contained in the test object.

There is a large amount of work on recognizing objects in a three-dimensional point cloud obtained using LiDAR and a stereo camera using a combination of various individual features and descriptors with classification by machine learning methods [21,22,23,24]. Semantic segmentation methods are also widely used, where structured classifiers are used instead of separate classifiers. Unlike the above approaches, architecture learns to extract features and classify objects from “raw” 3D data. The volumetric representation is also better than point clouds in the way it distinguishes free space from the unknown. In addition, methods using point clouds require point neighborhoods for calculations, which often becomes computationally unsolvable with a large number of points.

Inspired by the successful application of convolutional neural networks to solve recognition problems on two-dimensional images, some authors have expanded their use for stereo data. Such approaches treat the channel with “depth” as an additional channel, along with the usual channels R, G, B; however, geometric information in three-dimensional data are not fully used, which makes integration between visual points difficult.

For LiDAR data, features [23] have been proposed, locally obtained on data with a 2.5D representation, and some works investigate this approach in combination with a kind of so-called unsupervised learning [25]. In [24], an encoding is proposed that effectively uses depth information, but the approach is still two-dimensional-oriented; it produces a more accurate representation of the environment.

### 2.3. Existing Technologies

Currently, artificial intelligence for object recognition is being actively introduced into various fields of activity. Among all neural networks, CNN (convolutional neural networks) copes with this task, it made it possible to make a leap in the field of computer vision. The most current and widespread CNN architecture is aimed at two-dimensional images; there are also models among this architecture applied to three-dimensional images.

With the improvement and growth of technologies, as well as the growth of LiDAR sensors, the recognition of three-dimensional objects has reached a new stage. At the moment, various methods are being developed for the classification and reconstruction of 3D objects, segmentation of 3D scenes. To begin with, we look at the methods used to recognize three-dimensional objects.

With the help of special devices, a kind of three-dimensional point cloud is built, which represents a three-dimensional scene. When implementing this method, the following problems may occur:Our cloud consists of a chaotic order of points;The relationship of points is a certain distance by which it becomes necessary to contact the network;Data loss.To solve these problems, you can use the following methods:Sorting. The method is not the most effective;Using a symmetric function to aggregate information. That is, a function whose value does not change depending on the order of the elements.

This method was implemented in PoinNet. The PointNet architecture contains three key modules: a maximum pooling layer as a symmetric function for aggregating information from all points, a local and global information combination structure, and two joint alignment networks that align input points and point functions.

The authors of the PointNet architecture propose to transmit information about points (x, y, z) directly to a deep neural network. In order to find the symmetry function for the disordered input, the general function defined in the current set is approximated by applying the symmetric function to the transformed elements. PointNet approaches the function of a multilayer perceptron network and has been transformed by the composition of a monotonous function and a maximum union function. The output of the function forms a vector, which is considered as the global signature of the input data set and is fed to each point, combining global objects with each of the point functions. Then, new functions at one point are extracted based on the combined point functions, since local as well as global information will be known at the point. The joint alignment network forming the third module is inspired by the fact that the semantic labeling of the current cloud should be non-invariant if the current cloud undergoes geometric transformations. PointNet predicts the affine transformation matrix by the T-net architecture and directly applies this transformation to the coordination of input points. T-net consists of point independent extraction functions, maximum pooling, and fully connected layers. The transformation matrix in the object space has a higher dimension. Thus, for optimization, an ordering term that restricts the function transformation matrix to be close to the orthogonal matrix is added to the softmax training loss. As a result, for each object (class) the points required for the assessment are given.

There is a problem that in three-dimensional scenes, no one has considered which data format is best suited for processing. For the point cloud, a voxel grid was used—this is the most intuitive way to embed 3D objects into a grid so that they look like pixel images. It can be obtained from a point cloud (image). To do this, you need to create a 32 × 32 × 32 array, which will be filled with zeros. Then it is necessary to calculate the points inside each small voxel, and also perform scaling. The voxel was then assigned a common color; moreover, it was also possible to use the arithmetic mean of the points to assign a color.

As a neural network working with a voxel grid, consider the three-dimensional convolutional neural network VoxNET for real-time object recognition. It was proposed by the authors as a technology for recognizing cars and pedestrians [26]. The architecture consists of 3 main parts -the Feature Learning Network, convolutional middle layers (Convolutional Middle Network) and the Region Proposal Network. Of course, in the presence of a complex Vox dataset, the grid will not be the best choice. VoxNET is a relatively old technology, however, without it, the review of existing solutions would not be complete.

All the technologies considered are not suitable for solving the task of restoring a realistic urban environment, due to their high computational complexity, a large number of images submitted to the input and the lack of automatic conversion of a point cloud into a mesh with optimization of the final model.

### 2.4. Problem Statement

The task of the work is to implement the recognition of vehicles and road infrastructure objects for the further creation of a simulator of the urban environment. At the input, the neural network receives a photo, at the output—the classification of the road infrastructure object and its mesh for loading into the simulator.

To solve this problem, it is convenient to use object-oriented, structural and functional programming. To display the results of the program and the ability to work with the program on an intuitive level, a convenient and intuitive interface is required, displaying all the necessary menus, hints, and so on.

Python was chosen to implement the system. Python is a modern object-oriented language with a large number of libraries available. The convenience and extensibility of the language can significantly simplify development, and the use of additional libraries allows you to use a lot of ready-made already-implemented functions.

## 3. Results

### 3.1. System Operation Design

We will carry out the design of the solution using the standard methodology of business process modeling IDEF0. A contextual diagram of the functioning of the entire system is shown in Figure 1.

Training and validation samples are submitted to the input of the system, as well as the dynamic libraries used are connected. Random images are taken as the basis for recognition.
At the output of the system, we get a recognized image;The mechanisms are the Python user and environment;The decomposition of the context diagram is shown in Figure 2.

According to the decomposition, data are first prepared in the system, training and validation samples are loaded, and dynamic libraries are connected; after that, a neural network is created. Then the model is launched. The final process is the recognition and evaluation of the model. Decomposition of the “Data Preparation” process is shown in Figure 3.

In the process of data preparation, a data source, dynamic libraries are connected, after which the data are prepared for processing. Decomposition of the “Model Startup” process is shown in Figure 4.

During the startup process, the model is loaded and trained based on a previously created neural network. Decomposition of the process “Work and evaluation of the model” is shown in Figure 5.

During the operation of the model, accuracy is first evaluated on two samples—training and validation; after that, the model is evaluated by parameters.

At the end of the process, randomly selected images are recognized in order to demonstrate the operation of the model. Recognition data are output to the user.

### 3.2. System Architecture

The application under development consists of a main unit (a neural network performing data recognition), a training module, an evaluation module for the resulting model, an initial set of images, and plugins. The architecture of the neural network used is shown in Figure 6.

First, libraries are connected, then images are loaded into the program and prepared for processing; after that, the neural network is trained, or the already trained model is loaded. After that, the correctness of the network operation is checked, and its operation is also checked on several randomly selected images.

### 3.3. Description of the Algorithm

In a three-dimensional convolution, the convolutional layer repeats the same operations as in a two-dimensional one. It’s just that this implementation is already happening on several pairs of two-dimensional matrices. The output features are a weighted sum of the input features (where the weights are the values of the kernel itself) and are located in approximately the same places as the output pixels on the input layer.

Whether the input features fall into “approximately the same place” or not depends on whether it is located in the region of the kernel that created the output signal; this means that the size of the kernel of a convolutional neural network determines the number of features that will be combined to produce new features at the output.

Each filter in the convolutional layer creates only one output channel, and does this by “sliding” each filter kernel through the corresponding input channels, creating a processed version of each channel. Some kernels may have more weight than others in order to focus more on certain input channels (for example, a filter may give the red channel of the kernel more weight than other channels, and thus respond better to differences in the image along the red channel).

Each processing version in the channels is then added together to form a channel. The kernel of each filter generates one version of each channel, and the filter as a whole creates a common output channel.

Finally, each output file has its own offset. The offset is added to the output channels to create the final output channel.

The result is the same for any number of filters: each filter uses its own set of kernels and scalar offsets to process the input signal, as described above, to create an output channel; they are then combined together to produce a common output, with the number of output channels equal to the number of filters; this usually involves applying non-linearity before transferring the input data to another convolutional layer, and then repeating the process.

At the initial stage, the neural network is untrained. In a general sense, training is understood as the sequential presentation of an image to the input of a neural network, from a training set, then the received response is compared with the desired output. Then this error delta must be extended to all connected neurons of the network. Thus, neural network training is reduced to minimizing the error function by adjusting the weights of synaptic connections between neurons. The error function refers to the difference between the received response and the desired one.

The error back propagation algorithm determines the strategy for selecting weights and network parameters using gradient optimization methods and is considered one of the most effective learning algorithms.

The method is used to minimize the neural network error and obtain the desired output. The main idea of the method is to propagate error signals from the network output to its inputs in the opposite direction of direct signal propagation.

The first stage of training is network initialization.

At the second stage of training, a training sample is presented and the values of the signals of the neurons of the network are calculated.

At the third stage, it is necessary to change the weight coefficients so as to minimize the objective function:(1)$$E=\frac{1}{2}(\sum_{j=1}^{k}y(x_{i})-d_{i}{)}^{2},$$

At each step of the training, the weight coefficients are changed according to the formula:(2)$$w_{ij}(t+1)=w_{ij}(t)-\alpha \gamma_{j}F^{\prime }(S_{j})y_{i},$$
where *α* is the learning coefficient, *t* and *t* + 1 are the moments of time before and after the change of weights and thresholds, respectively, the indices *i* and *j* denote the neurons of the first and second layer of neurons, respectively, *γ_j_* is the difference between the network output and the reference, *F* is the activation function.

The thresholds of the network are changed by the formula:(3)$$T_{j}(t+1)=T_{j}(t)+\alpha \gamma_{j}F^{\prime }(S_{j}),$$

The error for the hidden layer with index *i* is calculated through the errors of the next layer with index *j* as follows:(4)$$\gamma_{i}=\sum_{j}\gamma_{j}F^{\prime }(S_{j})w_{ij},$$

The learning process continues until the number of iterations exceeds the established limit [27].

### 3.4. Formation of a Training Sample

As a subject area, ground vehicles are selected, namely: cars, buses, trucks, motorcycles, bulldozers, excavators. A ready-made dataset providing sufficient variety and high accuracy of markup was not found, so it was decided to collect the dataset from various sources, including: the website auto.ru [28]. Images were downloaded from all ads of the first two pages of each of the target categories; after that, the data were filtered out of duplicates and inappropriate images (details, interior, etc.)

The following datasets were also used:«Is it a Train or Bus» [29], from here images with buses were used;«UK Truck Brands Dataset» [30], which was used to create a sample with trucks;«Vehicle Dataset» [31], from which images of motorcycles and cars were taken;«Open Images Dataset» [32]. Images with buses and trucks were used from this dataset.

We also collected our own data set, which was collected by means of an installation mounted on a car (Figure 7) that drove through the streets of Moscow [33,34].

The result is a dataset consisting of 3000 images. The entire sample was divided into 3 parts: training 70%, validation 10%, testing 20%. The distribution of the data contained in the dataset by class is shown in Figure 8.

The structure of the neural network includes convolutional and fully connected layers and layers of subdiscretization (max-pooling). An example of such an architecture is shown in Figure 9. To reduce the effect of retraining, batch normalization and dropout were used. As an activation function—ReLU [35], as the most common today.

The current architecture uses the following order. First, the features are extracted using five convolutional layers. Then, the last feature map is converted into a vector and fed to the input of a hidden fully connected layer, followed by an output layer with 6 neurons.

The cross-entropy was used as a loss function. Accuracy was also monitored during the training process. The values of metrics at the training and evaluation stages are shown in Figure 10.

Cross entropy-based loss function:(5)C=−(ylna+(1−y)ln(1−a))

If the output signal is close to the desired value, the value of the loss function is close to 0:(6)$$\underset{a\to y}{\lim}C=0$$

There is no slowing down of learning when $\sigma^{\prime }(z)\to 0$
(7)$$\frac{\partial C}{\partial w_{i}}=x_{i}\cdot (\sigma (z)-y)$$
(8)$$\frac{\partial C}{\partial b}=\sigma (z)-y$$

During the training, the following hyperparameters were exposed: the learning rate is 0.001, the size of the batch is 32, the maximum number of epochs is 1000. At the same time, feedback was used for an early stop in order to stop the training of the model in the absence of improvements in the loss function on the validation part of the sample. The model was trained for 41 epochs, after which an early stop occurred, since the highest indicators were at 31 epochs.

As a result, 96.4% accuracy was achieved on the training part of the data block, 88.2% on the test sample and 90.3% on the validation.

An error matrix was also visualized to assess which classes the model makes the most mistakes on (Figure 11). The error matrix is a tabular distribution between the correct answer and the one given by the neural network. According to the rows, the correct answer, according to the columns– is the one given by the neural network. The number of matches at intersections. The correct answers will lie on the main diagonal of the array.

It can be seen from the error matrix that the excavator is predicted extremely inaccurately, while there are no errors at all for bulldozers; this may be due to the insufficient number of examples of this class in the sample.

For comparison with already existing solutions, the networks YOLOv5 [37], Mask R-CNN [38], ResNeXt [39], VGG16 [40] were selected; they were trained on the same data sets as the network we are developing, namely: a set of data that was collected from the site “auto.ru”, “It is Train or Bus”, “UK Truck Brands Dataset”, “Vehicle Dataset”, “Open Image Dataset”, as well as the collected dataset with streets of Moscow.

Comparisons of classification quality for each class are presented in Table 1, Table 2, Table 3 and Table 4. The best indicators for each class are highlighted in bold.

As can be seen from the classification results, the dredge class shows the worst results; this is due to the fact that the training was conducted on a small sample of data; it is planned to finalize this in the future.

Training schedules for YOLOv5, Mask R-CNN, ResNeXt and VGG16 networks are shown in Figure 12, Figure 13, Figure 14 and Figure 15.

In order to choose the optimal training period for YOLOv5, in order to avoid a situation not before training or retraining the model, it was decided to train the model on a different number of epochs and compare the results obtained. Comparison of training in 25, 50 and 75 epochs and their visualization is carried out using TensorFlow and is shown in Figure 13.

Based on the comparison graphs of training in 25, 50 and 75 epochs, the best results were shown at 75 epochs, since the degree and correctness of detection using this model turned out to be much higher, and the error is much smaller compared to the rest.

To choose the optimal training period for Mask R-CNN, in order to avoid a situation not before training or retraining the model, it was decided to train the model on a different number of epochs and compare the results obtained. Comparison of training in 20, 40 and 60 epochs and their visualization is carried out using TensorFlow and is shown in Figure 14. 

Based on the comparison graphs of training in 20, 40 and 60 epochs, the best results were shown at 60 epochs, since the degree and correctness of detection using this model turned out to be much higher, and the error is much smaller compared to the rest.

To choose the optimal training period for ResNeXt, in order to avoid a situation not before training or retraining the model, it was decided to train the model on a different number of epochs and compare the results obtained. Comparison of training in 60, 75 and 90 epochs and their visualization is carried out using TensorFlow and is shown in Figure 15. 

Based on the comparison graphs of training in 60, 75 and 90 epochs, the best results were shown at 90 epochs, since the degree and correctness of detection using this model turned out to be much higher, and the error is much smaller compared to the rest.

To choose the optimal training period for VGG16, in order to avoid a situation not before training or retraining the model, it was decided to train the model on a different number of epochs and compare the results obtained. Comparison of training in 25, 50 and 75 epochs and their visualization is carried out using TensorFlow and is shown in Figure 16. 

Based on the comparison graphs of training in 25, 50 and 75 epochs, the best results were shown at 75 epochs, since the degree and correctness of detection using this model turned out to be much higher, and the error is much smaller compared to the rest.

After conducting a study and comparing the model we developed with existing popular solutions (YOLOv5, Mask R-CNN, ResNeXt, VGG16), we can draw the following conclusions:The network developed by us has the best classification time (5.01442 ms) among the considered models. The closest result was shown by YOLOv5 (5.48544 ms).The network developed by us has the best classification accuracy (88.2%) among the models considered. The closest result was shown by the Mask R-CNN model (88.19%).

Based on the demonstrated results, we can conclude that the model we have developed is best suited for solving the task at hand—recognition and classification of vehicles and road infrastructure objects for further creation of an urban environment simulator. Despite the fact that the average classification accuracy is not much higher than Mask R-CNN, however, the performance increases.

## 4. Conclusions

As a result of this study, the following results were obtained:The system was designed using the standard methodology of business process modeling IDEF0.The system architecture has been developed.A data set has been formed, consisting of data from open sources (data collected from the site “auto.ru “, datasets: “It is Train or Bus”, “UK Truck Brands Dataset”, “Vehicle Dataset”, “Open Image Dataset”), as well as data collected through the installation, fixed on a car that drove through the streets of Moscow.The study and comparison of existing popular neural network models that are used for similar tasks, namely—YOLOv5, Mask R-CNN, ResNeXt, VGG16; these models were trained on the same data as the model being developed.

## Figures and Tables

**Figure 1 sensors-22-05199-f001:**
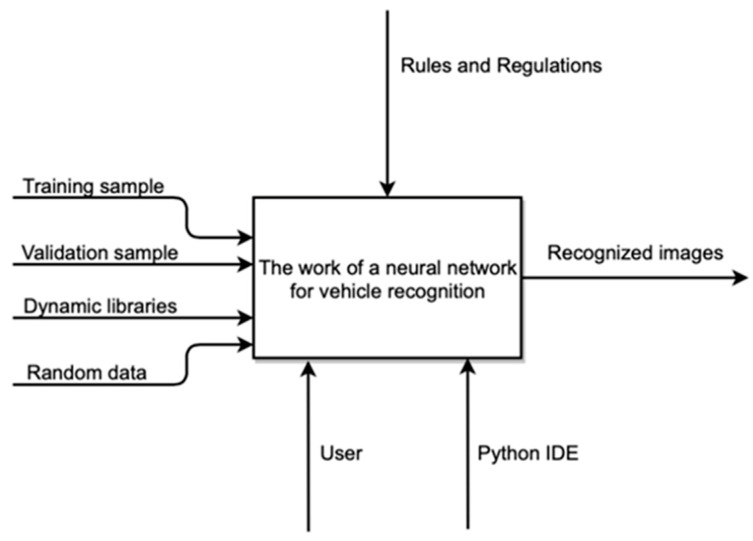
Contextual diagram of the system functioning.

**Figure 2 sensors-22-05199-f002:**
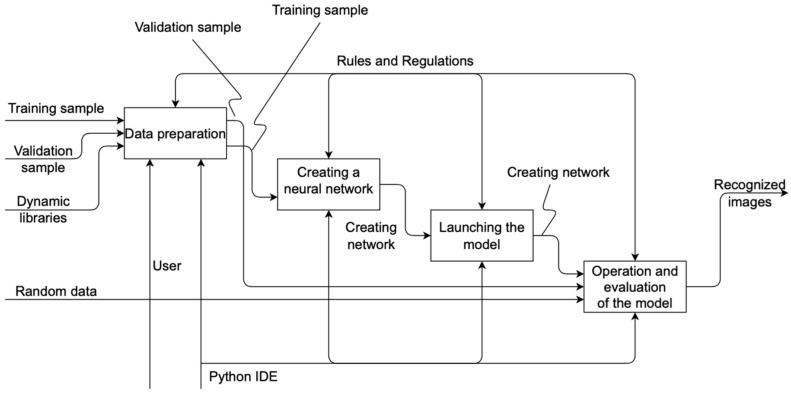
Decomposition of the context diagram.

**Figure 3 sensors-22-05199-f003:**
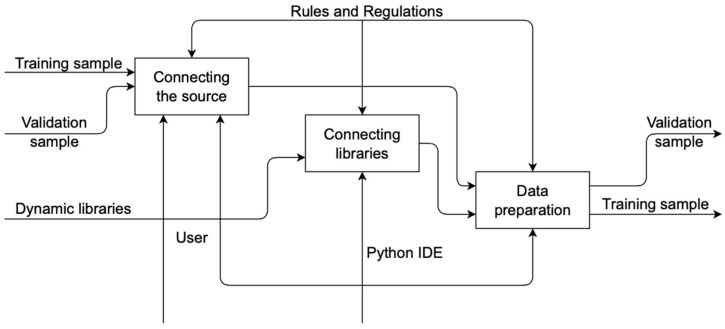
Decomposition of the “Data Preparation” process.

**Figure 4 sensors-22-05199-f004:**
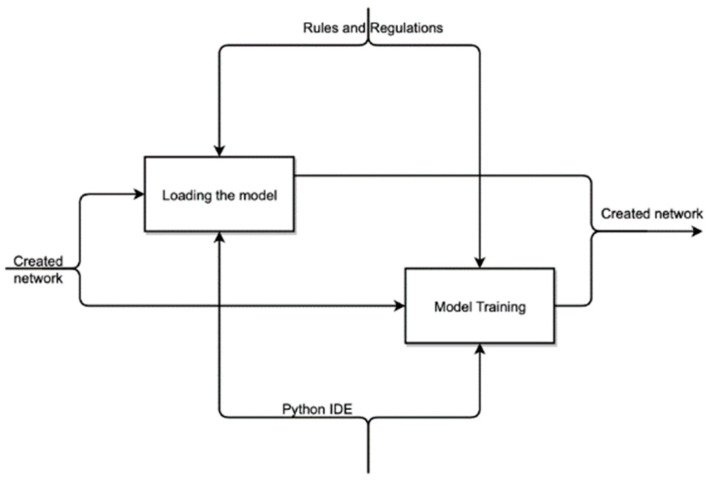
Decomposition of the “Model Startup” process.

**Figure 5 sensors-22-05199-f005:**
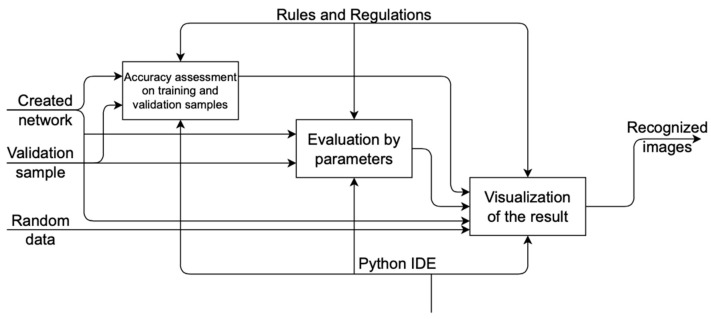
Decomposition of the process “Work and evaluation of the model”.

**Figure 6 sensors-22-05199-f006:**
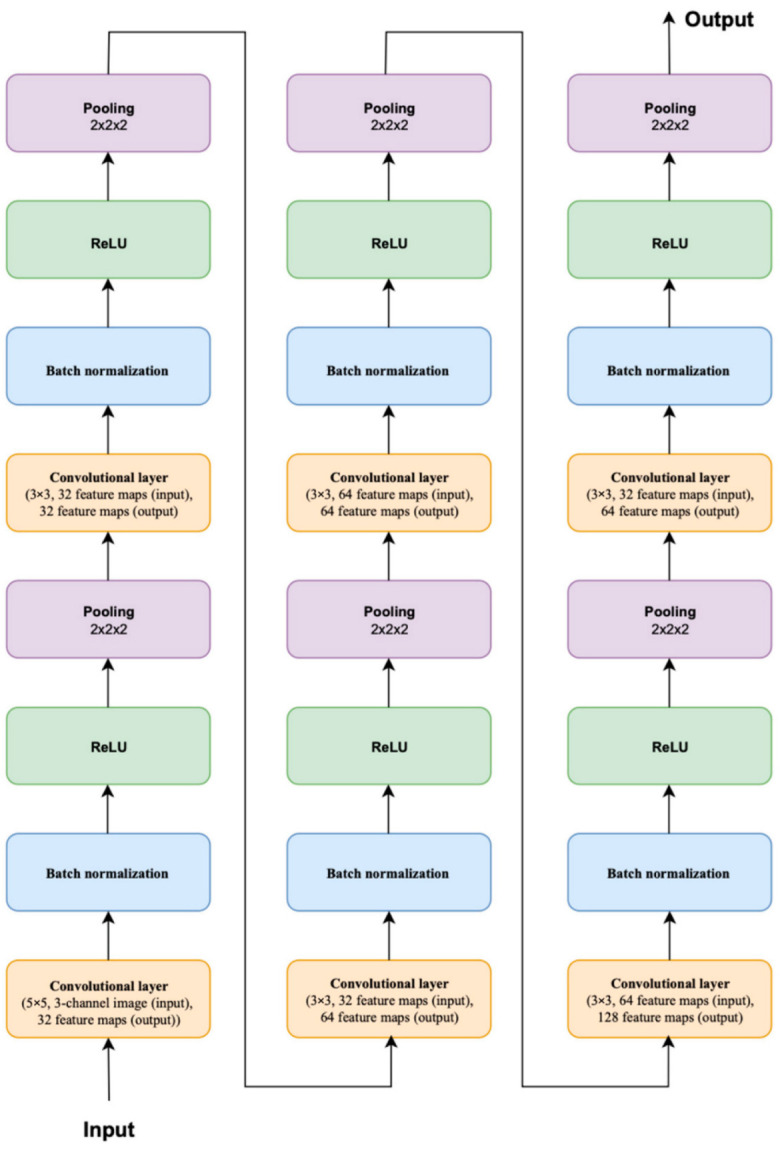
Neural network architecture.

**Figure 7 sensors-22-05199-f007:**
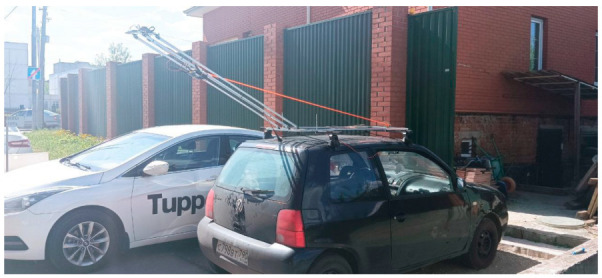
Data Collection Installation.

**Figure 8 sensors-22-05199-f008:**
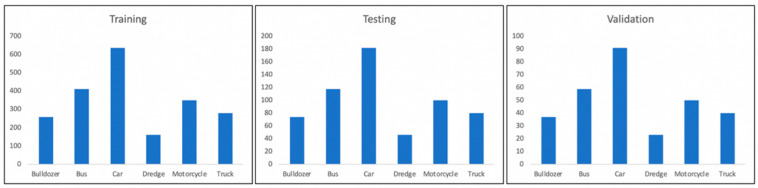
Parameters of the training, testing and validation samples.

**Figure 9 sensors-22-05199-f009:**
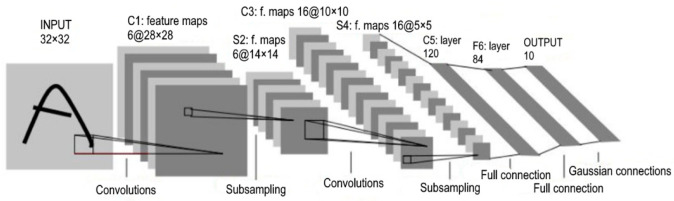
Example architecture of a convolutional network [36].

**Figure 10 sensors-22-05199-f010:**
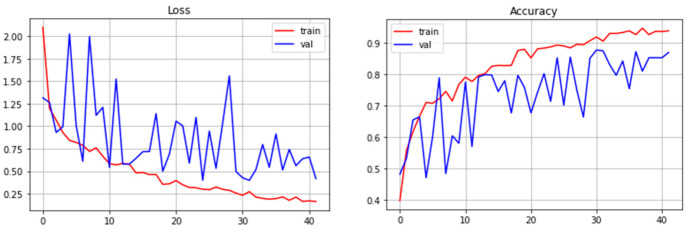
Graphs showing the learning process.

**Figure 11 sensors-22-05199-f011:**
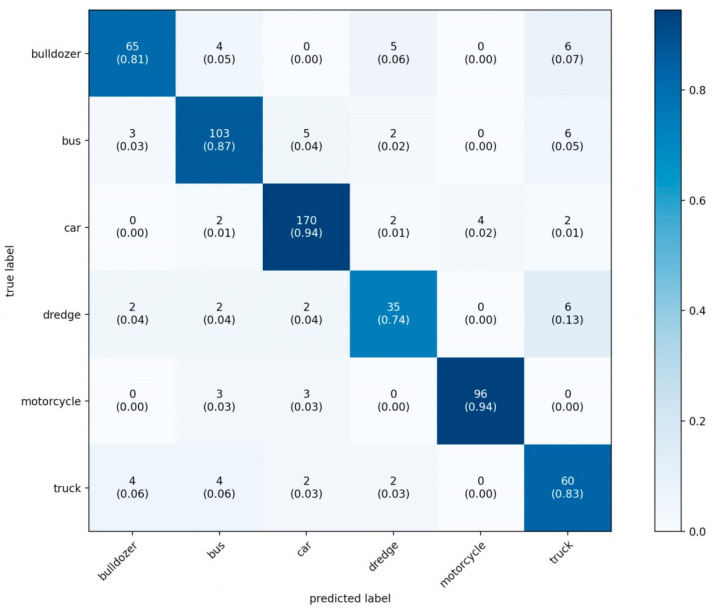
Visualization matrix of errors (testing sample).

**Figure 12 sensors-22-05199-f012:**
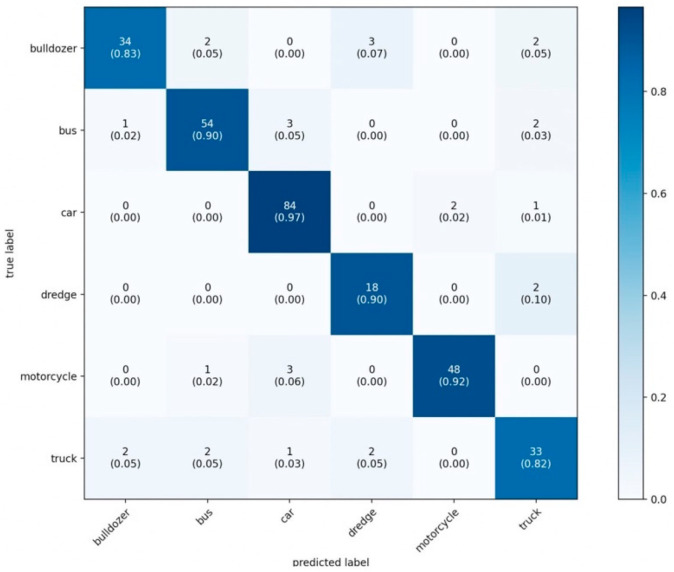
Visualization matrix of errors (validation sample).

**Figure 13 sensors-22-05199-f013:**
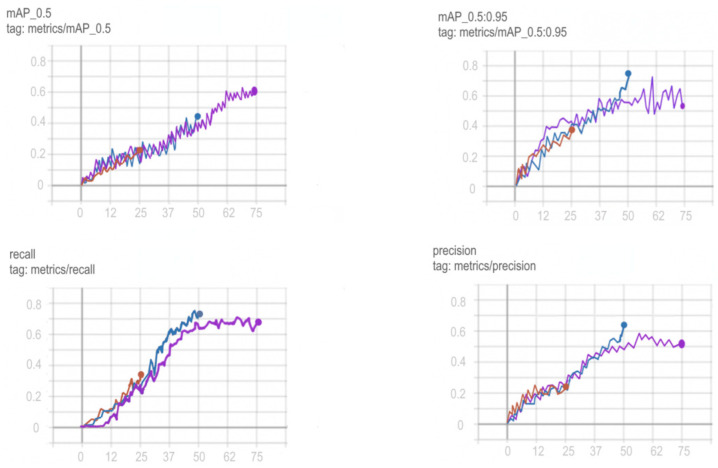
Comparison of training YOLOv5 at 25 (red graph), 50 (blue graph) and 75 (purple graph) epochs.

**Figure 14 sensors-22-05199-f014:**
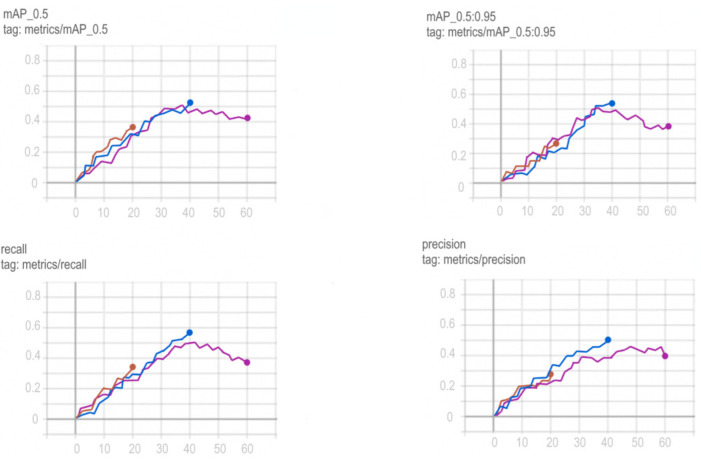
Comparison of training Mask R-CNN at 20 (red graph), 40 (blue graph) and 60 (purple graph) epochs.

**Figure 15 sensors-22-05199-f015:**
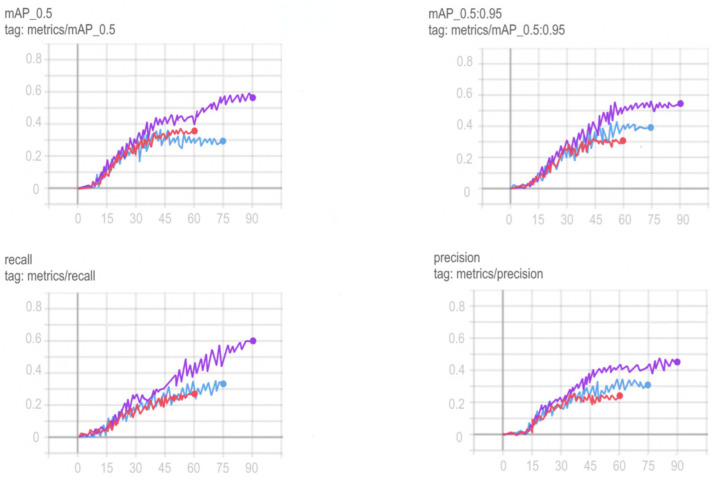
Comparison of training ResNeXt at 60 (red graph), 75 (blue graph) and 90 (purple graph) epochs.

**Figure 16 sensors-22-05199-f016:**
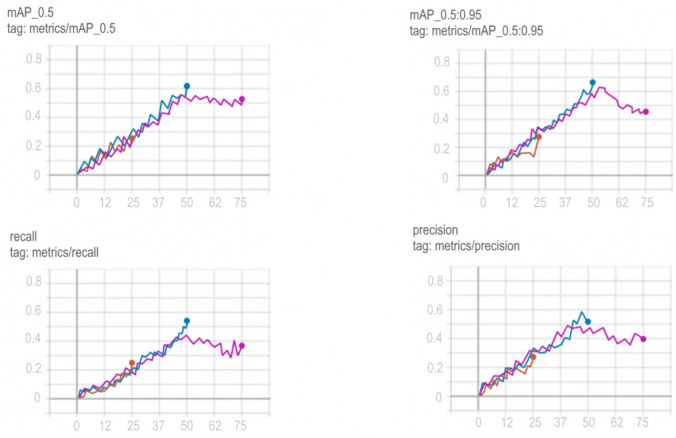
Comparison of training VGG16 at 25 (red graph), 50 (blue graph) and 75 (purple graph) epochs.

**Table 1 sensors-22-05199-t001:** Comparison of precision grades for each class. The best indicators for each class are highlighted in bold.

Vehicle	Our Net	YOLOv5	Mask R-CNN	ResNeXt	VGG16
Bulldozer	**0.87**	0.71	**0.87**	0.71	0.71
Bus	0.83	0.63	**0.86**	0.65	0.67
Car	**0.89**	0.63	0.86	0.69	0.67
Dredge	**1.00**	0.77	0.92	0.79	0.77
Motorcycle	0.92	0.67	**1.00**	0.77	0.91
Truck	0.84	0.83	0.86	0.83	**0.91**

**Table 2 sensors-22-05199-t002:** Comparison of recall scores for each class. The best indicators for each class are highlighted in bold.

Vehicle	Our Net	YOLOv5	Mask R-CNN	ResNeXt	VGG16
Bulldozer	**1.00**	0.67	0.93	0.71	0.77
Bus	**0.89**	0.71	0.86	0.79	0.77
Car	0.91	0.91	**0.92**	**0.92**	0.91
Dredge	0.29	0.63	**0.92**	0.69	0.77
Motorcycle	**0.92**	0.67	0.75	0.67	0.67
Truck	0.85	0.67	**1.00**	0.67	0.71

**Table 3 sensors-22-05199-t003:** Comparison of F1 scores for each class. The best indicators for each class are highlighted in bold.

Vehicle	Our Net	YOLOv5	Mask R-CNN	ResNeXt	VGG16
Bulldozer	**0.93**	0.69	0.9	0.71	0.74
Bus	**0.86**	0.67	**0.86**	0.71	0.71
Car	**0.90**	0.74	0.89	0.79	0.77
Dredge	0.44	0.69	**0.92**	0.73	0.77
Motorcycle	**0.92**	0.67	0.86	0.71	0.77
Truck	0.85	0.74	**0.92**	0.74	0.80

**Table 4 sensors-22-05199-t004:** Comparison of time and accuracy of classification of different networks. The best indicators for each class are highlighted in bold.

Net Name	Accuracy	Time
Our net	**88.20%**	**5.01442 ms**
YOLOv5	69.37%	5.48544 ms
Mask R-CNN	88.19%	6.76321 ms
ResNeXt	73.49%	6.97524 ms
VGG16	75.82%	5.60733 ms

## Data Availability

Not applicable.

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
