# Peer review of "On the Problem of Restoring and Classifying a 3D Object in Creating a Simulator of a Realistic Urban Environment"

_sensors, 2022, doi:10.3390/s22145199_

Round 1
Reviewer 1 Report
The auther basically solves my questions, so i suggest accepting this paper.
Reviewer 2 Report
The authors responded to all comments.
This manuscript is a resubmission of an earlier submission. The following is a list of the peer review reports and author responses from that submission.
Round 1
Reviewer 1 Report
The authors propose a vehicle recognition system using a convolutional neural network. The idea is interesting but the authors do not compare themselves with the studies already existing in this field. Moreover, some adjustments must be made to the paper.
- Authors should extend the abstract by briefly describing the application context, the rationale for their proposal, and the results (from the perspective of classification accuracy) obtained.
- On line 28, write "patterns" and not "pat-terns".
- The authors should extend the introduction by including concepts related to the application context of the study they propose. There is a need to conduct a state-of-the-art study on the topic of vehicle recognition in order to better define the contribution the authors make with their study. The authors should also include The authors should also include appropriate bibliographic references in order to give scientific weight to what they describe.
- Also in the introduction, at the end, authors should include a brief description about the chapters that are to follow.
- Authors should include a literature reference regarding the "three-dimensional Hough method," in line 143.
- The authors illustrate the "tasks of image classification" on page 7, inserting in the bulleted lists: Input, training, validation; but the testing phase of the trained network is missing. In particular, the testing phase of the network should be carried out with a dataset completely unknown to the network in order to enhance the classification performances.
- Regarding the paragraph "Formation of a training sample" the authors should form a third dataset, the testing dataset, containing images completely unknown to the network and evaluate the accuracy, precision and re call on that dataset.
- Authors should, in addition, include the equations for the accuracy, re-call, and precision parameters, and the confusion matrix indicating the classification accuracy for each class.
- The authors should include a "Discussion" chapter in which they compare their study (and thus the classification results, the dataset, the neural network) with those proposed and existing in the state of the art. We should emphasize, the work proposed in this study and the innovative contribution.
- Approximately half of the bibliographic references included in the study appear to be somewhat dated, from 6/7 years ago. The authors, in addition to extending the bibliography by doing some state of the art, should consider more modern references (from the last 4 years maximum).
Reviewer 2 Report
The writing of this paper is confusing. The research is inadequate and the methods used in this paper are outdated. This paper is not suitable for publication in this journal.
- Line 28, “pat-terns”, hyphens are unnecessary.
- “But in autonomous operation, neural networks do not provide the necessary functionality.” This expression is not accurate.
- Lines 45-85 can be remove, or briefly summarized in 1-2 sentences.
- “Among all neural networks, CNN (convolutional neural networks) copes with this task, it made it possible to make a leap in the field of computer vision. ” This sentence needs to be corrected.
- Lines 252-253, ”As a neural network working with a voxel grid, consider the three-dimensional convolutional neural network VoxNET for real-time object recognition”. There are many more sentences like this in the text that need to be corrected.
- Lines 334-362 can be briefly expressed as 1-2 sentences.
- Lines 272 and 273 can be merged into one paragraph. There are many more strange subparagraphs like this one in this paper.
- It is recommended that the abstract and introduction of this paper should be rewritten to highlight the contribution of the article.
- List the relevant recent works.
- The paper lacks comparative experiments with related methods.
Round 2
Reviewer 1 Report
The authors have improved the paper, but some critical issues remain. The following will be reported.
1. The abstract was not extended according to the suggestion made in the previous review.
2. The introduction should be extended as much as possible; also introducing the appropriate bibliographical references. In addition, the comment from the previous round was not fully satisfied; it is not apparent at the end of the introduction a description of the chapters the authors are going to cover.
3. The new version fails to evince the addition of the bibliographic reference for "three dimensional Hough method," as suggested in the previous round.
4. Items 6 and 7 of the previous review were not satisfactorily answered. In fact, the authors did not remodel the tests by splitting the database properly, i.e., into 3 different datasets: training, validation and testing and reevaluate the performance by applying the testing dataset to the neural network.
5. The authors left out the figure related to the error matrix, but generally an appropriate way to emphasize performance is to include the confusion matrix that shows the network's accuracy at classifying the input data into the correct classes
Author Response
Thanks for the comments provided!
1. The abstract was not extended according to the suggestion made in the previous review.
>The abstract has been expanded in accordance with the comments. A description of the essence of the article was added, what was done to achieve the tasks set and where the result will be used in the future. (Lines 9-34)
2. The introduction should be extended as much as possible; also introducing the appropriate bibliographical references. In addition, the comment from the previous round was not fully satisfied; it is not apparent at the end of the introduction a description of the chapters the authors are going to cover.
>After analyzing publications on similar topics in Sensors, it was decided to expand the "Introduction" section by merging it with the "Related work" section, adding appropriate bibliographic references to the existing systems under consideration. Also, a description of the subsequent chapters of the article was added to the end of the "Introduction" section. (Lines 207-216)
3. The new version fails to evince the addition of the bibliographic reference for "three dimensional Hough method," as suggested in the previous round.
>A bibliographic reference to the three-dimensional Hough Transform has been added [21, 22] (line 253)
4. Items 6 and 7 of the previous review were not satisfactorily answered. In fact, the authors did not remodel the tests by splitting the database properly, i.e., into 3 different datasets: training, validation and testing and reevaluate the performance by applying the testing dataset to the neural network.
>The data set has been supplemented (in addition to datasets from open sources, data collected through a developed installation mounted on a car is used). The dataset was divided into 3 samples – training, test and validation, which were divided in the ratio – 70%/20%/10%, respectively. (Lines 510-513)
5. The authors left out the figure related to the error matrix, but generally an appropriate way to emphasize performance is to include the confusion matrix that shows the network's accuracy at classifying the input data into the correct classes
>The figure "Error Matrix" was moved to the part describing the training of the developed network. (Line 541).
Reviewer 2 Report
In the section “3 Results”,there is still no added comparison experiments with related methods. The classification networks used in this paper is so old. And there is no explanation as to why such an old model was chosen and used in this paper.
Author Response
Thanks for the comments provided!
In the section “3 Results”,there is still no added comparison experiments with related methods. The classification networks used in this paper is so old. And there is no explanation as to why such an old model was chosen and used in this paper.
>In the "Results" section, a comparison was added with popular neural network models that are used for similar tasks, namely – YOLO, Mask R-CNN, ResNeXt, VGG. These models were trained on the same data as the model being developed. Comparisons are presented in the form of tables. The conducted comparisons help to justify the use of the developed network model for further use when creating a simulator of the urban environment.
It is worth noting that this article does not use the VoxNET architecture, which is also being considered, but uses the architecture developed by us. VoxNET is a relatively old technology, but without it, an overview of existing solutions would be incomplete.
Round 3
Reviewer 1 Report
The authors have responded to most of the comments and improved the paper significantly. The only remaining doubt is in the experimental results section.
In fact, it seems that the values and results of the error matrix remained unchanged from the previous version. Given that the dataset has been extended and has been divided into three different datasets (70% training, 10% validation and 20% testing) one would expect different values within the error matrix. Counting all values within the error matrix should lead back to the number of elements in the testing dataset; instead, the value obtained (357) does not correspond to 20% of the testing dataset described in the paper. The authors should update and describe this section more clearly, correctly and comprehensively.
Reviewer 2 Report
1. The text in Figure 2,3,4,5 is unclear and a larger font size is recommended.
2. Lines 477-488, why do you split each sentence into a paragraph?
3. Lines 361-390, it is unnecessary to the features of pytorch in such detail.
4. It is better to express the loss function mathematically.
5. YOLO-V1 is used for object detection task. And Mask R-CNN is used for segmentation task. I wonder why you compare your method with these two models and how do you compare with them. Please give details.